# OpenReview forum: "Reasoning over Boundaries: Enhancing Specification Alignment via Test-time Delibration"
_ICLR.cc/2026/Conference — Submitted to ICLR 2026_

### Official Review · Reviewer_8wVs · 2025-10-29

**Soundness:** 2
**Presentation:** 3
**Contribution:** 3
**Rating:** 4
**Confidence:** 4

**Summary:**

This paper tackles the specification alignment task, which aims at making LLMs adhere to dynamic, scenario-specific rules (specs) for both safety and behavior. It introduced ALIGN3, a lightweight test-time deliberation method that uses a 3-step reasoning process to navigate these boundaries. To evaluate this, the authors present SpecBench, a benchmark spanning five real-world scenarios. Experiments on 33 models demonstrate that test-time reasoning significantly boosts alignment.

**Strengths:**

+ focus on a practical task
+ good performance
+ sound model architecture

**Weaknesses:**

1. evaluation bias:

For the evaluation, the authors used GPT-4.1 as the primary evaluator, while validated by a human study, which could be a potential source of bias. Models architecturally or functionally similar to GPT-4 might be unfairly favored. I recommend supplementing the LLM-based evaluation with a human-annotated gold standard, i.e., the authors can employ multiple human annotators to provide judgments and which can serve as an unbiased benchmark.


2. lack of a systematic, generalizable approach for creating a complete set of specs for a given domain:

The methodology for creating specifications is a notable weakness, as it relies on a manual process that is difficult to scale and validate. The approach of curating specs through human-expert and LLM collaboration is vulnerable to the creators' own biases and knowledge gaps. This is especially problematic for novel domains where both the human experts' understanding may be incomplete and the LLM's training data may be lacking. Consequently, the "completeness" of any resulting spec set is fundamentally unknown. More critically, the paper provides no systematic framework or clear stopping criteria for practitioners to create their own comprehensive specs in a new domain. This lack of a generalizable protocol limits the reproducibility and broader application of the otherwise valuable "specification alignment" paradigm.

3. limited addressment of dynamic and evolving specs:

In the introduction, the authors emphasized that real-world specifications are dynamic and evolve with changing user preferences and requirements. However, the proposed methods are fundamentally static, i.e.,  SpecBench is a fixed snapshot of 103 specifications. It does not provide a mechanism for testing a model's ability to adapt to new or modified specs after the initial evaluation. A model could be perfectly aligned with SpecBench's v1, but can fail completely on a later version, such as v2, where a key safety boundary has shifted. Also in ALIGN3, the TTD method required the specs to be provided at inference time. While this is more flexible than training-based methods, it does not address the core challenge of evolution. The method assumes the specs given are the correct, up-to-date set. It has no capability to reason about changes in specifications, handle conflicting rules from different versions, or identify when a spec is outdated.

**Questions:**

How do you ensure you haven't missed a critical safety risk or behavioral nuance? What is the stopping criterion?

---

> ### Author Response · Authors · 2025-11-20
> **Response to Reviewer 8wVs (1 / 4)**
>
> Thanks for your valuable insights. However, it seems there may be some misunderstanding regarding our proposed specification alignment paradigm. As noted in Sec. 1, “LLMs are expected to follow scenario-specific specifications spec set by individuals, companies, or organizations” and “we introduce specification alignment, the challenge of enabling LLMs to meet dynamic, fine-grained, and scenario-specific spec”. To avoid further confusion, we would like to clarify three key points before addressing your questions:
>
> 1. **A specification is a direct statement of user intent, not an attempt to enumerate all possible rules.** In our framework, a spec captures what an individual or organization needs in a specific context. It is not required to be globally exhaustive. Its content is inherently subjective because users understand their own constraints and priorities better than any model. They can define what matters most, set priority orderings, or specify how to handle conflicts. SpecBench already includes such conditional forms of specifications, for instance, the Health and Travel scenario contains several “if-then” behavioral spec that mirror realistic conditional requirements (Appendix J). This design prevents models from making implicit choices on behalf of the user, which neither TTD methods nor any model should attempt to do.
> 2. **Specification alignment is designed to operate naturally with changing and evolving requirements.** Real-world requirements shift as organizations, workflows, and societal conditions change. In our paradigm, users simply update the spec text to reflect their new intentions. Because Align3 and the TTD inference procedure rely solely on the specification given at inference time, they immediately adapt to newly added or revised requirements without any model level modifications. This makes the approach well suited for settings where specifications evolve continuously.
> 3. **The challenge lies in constructing evaluation scenarios, not in writing specifications.** SpecBench aims to evaluate and identify models or methods that can adapt to different given specifications after deployment. Writing a spec is usually straightforward for users, but constructing scenarios that effectively test a model’s ability to handle diverse, unfamiliar, or demanding requirements is the real challenge. SpecBench addresses this by providing varied and contrasting scenarios that expose how reliably a model can follow different behavioral and safety specifications.

---

> > ### Author Response · Authors · 2025-11-20
> > **Response to Reviewer 8wVs (2 / 4)**
> >
> > > **Q1. Evaluation Bias. (the 1st proposed weakness)**
> > >
> >
> > A1. We completely agree your concern, but this challenge is indeed shared by many current benchmarks:
> >
> > 1. **LLM-as-a-judge is widely adopted across mainstream benchmarks.** Beyond tasks with verifiable answers such as math reasoning, many general-purpose benchmarks (e.g., MT-Bench, ArenaHard [1,2]), safety-oriented benchmarks (e.g., AIR-Bench 2024 [3]), and even Google’s recent IMO-Proof-Bench for math reasoning [4] rely on LLM-as-a-judge. Given the scale of modern evaluation workloads, relying solely on human annotation is infeasible and prohibitively time-consuming and expensive for most research settings.
> > 2. **We take multiple steps to minimize evaluator bias.** We compare GPT-4.1 with human experts and with Qwen3-32B-thinking and find that both the agreement and the ranking trends remain highly consistent. Following prior work [3,5,6], our human study (Appendix F) shows a kappa of 0.84 across three annotators and GPT-4.1. Besides, the strong agreement and consistent rankings between GPT-4.1 and Qwen3-32B-thinking (Sec. 4.1 and Appendix G.6) also provide a cost-effective and locally deployable alternative for development.
> > 3. **A full human gold standard is infeasible and becomes outdated when specifications evolve.** Even our subset of 6180 judgments required about a week of effort from three annotators. Scaling this to all 33 models would require more than one million detailed judgments, which is not practical. More importantly, specifications change as user needs evolve, so any static gold standard quickly loses validity. Validating consistency with human experts on a representative subset (Appendix F) is therefore the most realistic and robust approach.
> > 4. **To further address your concern, we additionally test alternative evaluators in Sec. 6 of the revised version.** We incorporate the newest Claude-4.5-Haiku and DeepSeek-V3.2-Exp as evaluators for a subset of models and find that the ranking results remain largely stable. We also compute Kendall’s coefficient of concordance across all evaluators, obtaining a near-perfect $W=0.9948$ with $\chi^2=43.77$ (df = 11) and $p=7.97\times10^{-6}$, confirming extremely strong overall agreement. **Please refer to Sec. 6 in the revised version for more details.**
> >
> > [1] Judging LLM-as-a-Judge with MT-Bench and Chatbot Arena.
> >
> > [2] Length-Controlled AlpacaEval: A Simple Way to Debias Automatic Evaluators.
> >
> > [3] AIR-Bench 2024: A Safety Benchmark Based on Risk Categories from Regulations and Policies.
> >
> > [4] Towards Robust Mathematical Reasoning
> >
> > [5] Fofo: A benchmark to evaluate llms’ format-following capability
> >
> > [6] SALAD-Bench: A Hierarchical and Comprehensive Safety Benchmark for Large Language Models

---

> > > ### Author Response · Authors · 2025-11-20
> > > **Response to Reviewer 8wVs (3 / 4)**
> > >
> > > > **Q2. Lack of a systematic, generalizable approach for creating a complete set of specs for a given domain. (the 2nd proposed weakness)**
> > > >
> > >
> > > A2. We address your concern from the following perspectives:
> > >
> > > 1. **A specification is a direct expression of user intent and does not need to be exhaustive.** In our framework, a spec reflects what an individual or organization requires in a specific context. It is not meant to cover all possibilities. Users understand their priorities and constraints better than any model, and they can define what matters most, set priority orders, or specify how to resolve conflicts.
> > > 2. **We already provide representative guidelines for constructing specifications. Appendix D.5** presents clear rules for building behavioral specifications. Safety specifications are usually more direct since they reflect concrete constraints and safety boundaries of a scenario, and users can tailor them to their own context without much difficulty. Besides, **Appendix D.4** explains how human experts should supervise the construction process. We believe these guidelines are sufficient because real applications vary widely. Some users may only need a few simple specifications, while internal or sensitive settings may require more detailed and stricter rules. Overly rigid templates would limit users’ ability to express their actual needs.
> > > 3. **Specification alignment is inherently tailored to user preferences rather than designed to be globally unbiased.** As stated in Sec. 1, scenario-specific requirements are created by different individuals, institutions, or companies. Our goal is to align the model with the actual preferences of the user, not to introduce additional rules that the user did not specify. Insisting on a universal definition of completeness is unnecessary, as it may obscure the user’s true intent. Furthermore, asking an LLM or AI system to infer or supplement missing specifications falls outside the scope of our work.
> > > 4. **We recognize that providing high-level guidance for constructing specifications and data in new scenarios is helpful for practical deployment, and we include this in Appendix H of the revised version.** Appendix H provides an overarching instruction for designing specifications and data in new contexts. Since user roles, needs, and application contexts vary widely, the instruction is intentionally high level to offer sufficient flexibility for customization. More details are available in Appendix H.
> > >
> > > ---
> > >
> > > > **Q3. Limited addressment of dynamic and evolving specs. (the 3rd proposed weakness)**
> > > >
> > >
> > > A3. Thanks for your insights. We address your concern from the following perspectives:
> > >
> > > 1. **SpecBench serves not only as an evaluation dataset but also as a representative framework and workflow.** Beyond providing a benchmark, SpecBench offers a conceptual and practical workflow that individuals, institutions, and companies can follow to design their own specifications and data for internal, private, or lightweight use cases. As you noted, real-world specifications evolve with user preferences and requirements, and our proposed specification alignment paradigm is intentionally designed to support such real-world practices by illustrating how specifications can be structured and applied.
> > > 2. **SpecBench is designed to evaluate a model’s adaptability to scenario-specific specifications so that this capability can generalize to diverse future contexts, even though the current release is a static snapshot similar to existing benchmarks.**
> > >
> > >     We acknowledge that the present version of SpecBench reflects a fixed snapshot, which is consistent with the practice of existing benchmarks that rely on static datasets. Our goal, however, is to evaluate how effectively a model adapts to different specifications, enabling this adaptability to extend to newly emerging or evolving scenarios. As part of future work, we will update SpecBench periodically to broaden coverage and integrate representative new scenarios, thereby strengthening support for evolving requirements.
> > >
> > > 3. **We further validate our paradigm by simulating an evolving-spec setting in Appendix L of the revised version.**
> > >
> > >     To directly address your concern, we update the full specification sets of 5 scenarios and re-evaluate 12 selected models using the revised specifications, as detailed in Appendix L. The evaluation rankings under the original and updated versions remain highly consistent, with strong rank correlations (high ρ and τ with low p values below 10⁻⁴). This demonstrates that our specification alignment paradigm naturally accommodates evolving requirements and that SpecBench provides representative and reliable extrapolation ability for assessing model behavior under changing specifications. Please refer to Appendix L for full results.

---

> > > > ### Author Response · Authors · 2025-11-20
> > > > **Response to Reviewer 8wVs (4 / 4)**
> > > >
> > > > > **Q4. How do you ensure you haven't missed a critical safety risk or behavioral nuance? What is the stopping criterion?**
> > > > >
> > > >
> > > > A4. As discussed above, **a specification is a direct statement of user intent rather than an attempt to cover every possible rule, and it does not need to be exhaustive.** In our framework, a spec reflects what an individual or organization requires in a specific context. Its content is naturally subjective because users understand their own constraints, safety boundaries, and behavioral expectations better than any model. The level of detail and completeness depends on the user’s actual needs rather than on any universal standard.
> > > >
> > > > In practice, this means that different users will adopt different stopping criteria. Lightweight scenarios may only require a few straightforward specifications to express core constraints, while organizational or sensitive settings may call for more detailed and stringent specifications. What matters is that users include the requirements they deem critical. Since the goal of specification alignment is to ensure that the model follows the user’s stated intent and preference, the completeness of a specification is defined by the user’s own judgment about what risks or nuances must be addressed.
> > > >
> > > > ---
> > > >
> > > > **We sincerely appreciate your time and effort in reviewing our submission and our response, and we are of course happy to address any additional questions or concerns you may have.**

---

> > > > > ### Author Response · Authors · 2025-11-25
> > > > > **Follow-up on Our Rebuttal Submission**
> > > > >
> > > > > Dear Reviewer,
> > > > >
> > > > > Since it has been five days since we submitted our rebuttal, we wanted to kindly follow up in case you have any further thoughts or questions regarding our submission.
> > > > >
> > > > > We truly hope our responses were helpful in addressing your comments. If there’s anything further you’d like to discuss or clarify, we would be very grateful to hear your feedback before the discussion period ends.
> > > > >
> > > > > Warm regards,
> > > > >
> > > > > The authors

---

> > > > > > ### Author Response · Authors · 2025-11-27
> > > > > > **Kind Follow-up on the Discussion of Our Rebuttal**
> > > > > >
> > > > > > Dear Reviewer 8wVs,
> > > > > >
> > > > > > Thank you for reviewing our work and providing feedback. We submitted our response to your comments a few days ago. Since the rebuttal window is closing, we would greatly appreciate it if you could share any thoughts so that we can continue the discussion and address any concerns you may have.
> > > > > >
> > > > > > In summary, to address your points, we have added new experiments and discussion as shown below, which are also integrated into the updated version.
> > > > > >
> > > > > > First, we clarify the core ideas of the specification alignment paradigm. **A specification in our framework is a direct expression of user intent rather than an attempt to enumerate all possible rules, and therefore does not need to be comprehensive (Q2 and Q4)**, especially in scenarios where users define their own requirements. Specification alignment is inherently user centered, and users can construct specifications based on their own priorities with the help of our guidelines. Neverthless, to address your concern, we also provide high-level instructions for creating new specifications and data (**Appendix H**).
> > > > > >
> > > > > > Secondly, for **Q1**, we have taken multiple steps to minimize evaluator bias and include the newest evaluators (**DeepSeek-V3.2-Exp and Claude-4.5-haiku, Sec. 6**), showing that different evaluators exhibit high correlation.
> > > > > >
> > > > > > Finally, for **Q3**, we show that specification alignment supports evolving requirements. Although SpecBench is a static snapshot, our simulation of updated specifications demonstrates that rankings remain consistent, confirming the robustness of the paradigm (**Appendix L**).
> > > > > >
> > > > > > We would appreciate it if you could kindly take a look at our response to your comments. If you have any further questions, we are happy to address them!
> > > > > >
> > > > > > Best regards,
> > > > > >
> > > > > > The authors

---

### Official Review · Reviewer_vrxC · 2025-10-31

**Soundness:** 3
**Presentation:** 4
**Contribution:** 3
**Rating:** 6
**Confidence:** 4

**Summary:**

The work addresses the challenge of specification alignment for large language models (LLMs)—enabling LLMs to follow dynamic, scenario-specific safety and behavioral specifications (spec). It introduces a lightweight test-time deliberation (TTD) method called ALIGN3 and a unified benchmark SPECBENCH to advance this goal.
Specification alignment requires LLMs to adhere to two types of specs: safety-spec (defining scenario-specific safety boundaries, e.g., prohibiting harmful content in child storytelling) and behavioral-spec (guiding helpfulness, e.g., format constraints for code). The paper highlights a key "safety-behavior trade-off": strengthening one dimension often weakens the other (e.g., over-refusal ensures safety but reduces helpfulness).
To solve this, ALIGN3 uses three-step hierarchical TTD: 1) Behavior Optimization (maximizing compliance with behavioral-spec), 2) Safety-Guided Refinement (adjusting reasoning to avoid safety violations), and 3) Holistic Specification Audit (full spec review before final output). This minimizes token overhead while balancing safety and helpfulness.
The paper also presents SPECBENCH, a benchmark covering 5 scenarios , 103 specs, and 1,500 prompts. It uses the Specification Alignment Rate (SAR) metric, which prioritizes safety and rewards helpfulness for safe outputs.
Experiments on 18 instruct and 15 reasoning models  show three key findings: 1) SPECBENCH effectively reveals alignment gaps (most models score <65% SAR); 2) TTD generally improves alignment; 3) ALIGN3 advances the safety-helpfulness trade-off—e.g., boosting Qwen3-14B’s SAR from 51.03% to 62.92% with minimal tokens, approaching GPT-4.1’s 69.20%.
Overall, the work demonstrates test-time deliberation’s value for real-world specification alignment, providing a benchmark and method to guide scenario-specific LLM optimization.

**Strengths:**

1. The paper formally defines "specification alignment" as a critical challenge, emphasizing LLMs’ need to balance scenario-specific safety-spec (defining risk boundaries) and behavioral-spec (guiding helpfulness). This addresses a key gap in existing work that often overlooks dynamic, scenario-tailored requirements, providing a clearer framework for real-world LLM deployment.

2. Innovative Lightweight TTD Method (ALIGN3): ALIGN3’s three-step hierarchical deliberation  effectively balances safety and helpfulness. Unlike multi-pass TTD baselines (e.g., Self-Refine) that incur high token overhead, ALIGN3 achieves substantial gains (e.g., 11.89% SAR boost for Qwen3-14B) with minimal tokens (~1800 per sample), making it efficient for practical use.

3. SPECBENCH unifies evaluation across 5 realistic scenarios , 103 specs, and 1500 prompts. Its novel metric (SAR) prioritizes safety while rewarding helpfulness, enabling accurate measurement of the safety-helpfulness trade-off—filling a void in fragmented existing benchmarks.

**Weaknesses:**

1. The paper primarily uses GPT-4.1 to calculate the Specification Alignment Rate (SAR), with Qwen3-32B-thinking as a substitute. It lacks validation with human evaluators across all scenarios—only a small-scale human study (300 samples) is conducted, and no cross-verification with other LLMs (e.g., Claude 3) is done. This may introduce evaluator bias, especially for subjective behavioral-spec compliance.

2. the benchmark focuses on 5 English-language scenarios  but ignores low-resource languages or niche domains . This limits the framework’s applicability in non-English or specialized real-world contexts where scenario-specific specs differ significantly.

3. While ALIGN3 shows short-term SAR gains (e.g., 11.89% for Qwen3-14B), the paper does not explore long-term performance—such as whether gains degrade with extended use, or if the method causes "over-alignment" .

**Questions:**

1. Your attack enhancement uses WildTeaming to rewrite unsafe prompts, but only verifies semantic preservation with Qwen3-32B-thinking and human review. Have you tested if the attacked prompts still effectively trigger model safety violations across more diverse models (e.g., closed-source ones like Claude 3)? And how do you ensure the attack tactics don’t introduce scenario-specific biases (e.g., overusing "novel writing" in the Child scenario)?


2. Qwen3-32B-thinking is proposed as a cost-effective evaluator alternative to GPT-4.1, but its scores are slightly lower. For scenarios with strict safety requirements (e.g., Biochem’s dual-use risks), have you validated if this score gap leads to misjudgments of critical safety violations?


3. ALIGN3’s three-step TTD improves SAR, but ablation shows omitting any step reduces performance. Have you explored why combining Step 1 (behavior optimization) and Step 2 (safety refinement) comes closest to full ALIGN3, and if adjusting the order of steps (e.g., safety first) would yield better results?

4. Most models score <65% SAR, with GPT-5-chat leading. Have you analyzed whether GPT-5-chat’s advantage stems from better handling of specific scenario types (e.g., Code’s vulnerability constraints) or a general edge in balancing safety-behavior trade-offs?

---

> ### Author Response · Authors · 2025-11-20
> **Response to Reviewer vrxC (1 / 4)**
>
> > **Q1. The paper primarily uses GPT-4.1 to calculate the Specification Alignment Rate (SAR), with Qwen3-32B-thinking as a substitute. It lacks validation with human evaluators across all scenarios—only a small-scale human study (300 samples) is conducted, and no cross-verification with other LLMs (e.g., Claude 3) is done. This may introduce evaluator bias, especially for subjective behavioral-spec compliance.**
> >
>
> A1. Thanks for your helpful suggestions. We address your concerns from two perspectives:
>
> 1. **For validation with human evaluators, our human study required substantial manual effort. Compared with prior work such as Fofo-Bench (100 annotations) [1], SALAD-Bench (458 questions) [2], and AIR-Bench (1716 QA pairs) [3], our evaluation is already reasonably comprehensive, with 300 samples and 6180 annotations.** As detailed in Appendix F, annotators must read each prompt and response, check every specification in the scenario, and follow the evaluation protocol in Sec. 3.3. This is a time-consuming process, and the annotators spent about a week completing it. Similar to prior work, these 300 samples are sufficient to validate the consistency between human annotators and the GPT-4.1 evaluator.
> 2. **For cross-verification with additional LLM evaluators, beyond GPT-4.1 and Qwen3-32B-thinking, we also included the newest Claude-4.5-Haiku and DeepSeek-V3.1-Exp on a subset of models in Sec. 6 of the revised version.** The ranking patterns remain largely stable across evaluators. We also compute Kendall’s coefficient of concordance across all evaluators, obtaining a near-perfect $W=0.9948$ with $\chi^2=43.77$ (df = 11) and $p=7.97\times10^{-6}$, confirming extremely strong overall agreement. **Please refer to Sec. 6 in the revised version for more details.**
>
> [1] Fofo: A benchmark to evaluate llms’ format-following capability
>
> [2] SALAD-Bench: A Hierarchical and Comprehensive Safety Benchmark for Large Language Models
>
> [3] AIR-Bench 2024: A Safety Benchmark Based on Risk Categories from Regulations and Policies
>
> ---
>
> > **Q2. SpecBench focuses on 5 English-language scenarios but ignores low-resource languages or niche domains.**
> >
>
> A2. Thanks for your suggestions. We fully understand your concern. As noted in Sec. 3.1, **“SpecBench provides a foundation for organizations to establish their own specification boundaries in real-world applications.”** What propose a general workflow for constructing specifications, building data, and evaluating models, which individuals, institutions, and companies can extend to more user-customized or even internal scenarios beyond the five we cover.
>
> However, we agree that low-resource languages and niche domains are very important. **Due to time constraints and the substantial cost of accessing real projects in these low-resource areas and hiring niche domain experts, we are currently not able to cover them.** We view this as an important and necessary direction for future work.
>
> To help address your concern, incorporating scenarios in low-resource languages or niche domains would require the following steps:
>
> 1. Collaborate with real projects in those areas, clarify their requirements and risks, and work with niche domain experts to understand what kind of behavior and safety guarantees are actually needed in practice.
> 2. Following the SpecBench construction workflow in Sec. 3, define scenario-specific specifications and collect prompts, then iteratively refine them with continuous expert review and human quality control (Appendix D.4), as well as a targeted human evaluation study (Appendix F).
> 3. Bring in models that are relevant for those languages or domains and evaluate them using both training-based methods and test-time deliberation approaches such as Align3, in order to select appropriate solutions for real deployments.
>
> **We have also added this detailed workflow guidance in Appendix H of the revised version, which we hope can serve as a foundation for future work and assist individuals, institutions, and companies with similar needs.**

---

> > ### Author Response · Authors · 2025-11-20
> > **Response to Reviewer vrxC (2 / 4)**
> >
> > > **Q3. While ALIGN3 shows short-term SAR gains (e.g., 11.89% for Qwen3-14B), the paper does not explore long-term performance—such as whether gains degrade with extended use, or if the method causes "over-alignment".**
> > >
> >
> > A3. Thanks for your suggestions. We have the following explanations and also a question for clarification:
> >
> > 1. **Align3 is a test-time deliberation method that provides short-term SAR gains without modifying model weights.** Therefore, we believe it does not affect long-term or extended use. Align3 is specifically designed as a simple and efficient option for the specification alignment task, which inherently requires incorporating specifications. Similar to prior instruction-following or safety work (see Sec. 7), specification alignment targets user-defined rules or specifications. For tasks unrelated to specifications, such as math benchmarks like IMO-Bench [1] or general instruction evaluation like AlpacaEval [2], Align3 is unnecessary and therefore has no extended-use effect.
> > 2. Regarding “over-alignment”, what we aim to align are scenario-specific specifications defined by behavioral and safety boundaries. This is different from “over-refusal”, because direct refusals cannot achieve a high SAR score, as noted in Appendix G.5. In addition, **our formulation of the safety-behavior trade-off (Sec. 2.1 lines 166 to 168, Sec. 4.2 lines 342 to 354, and the case study in Appendix G.5) directly prevents behavior that collapses into overly conservative behavior.**
> >
> > **However, we may still misunderstand what you precisely mean by “over-alignment”. Could you clarify the term so that we can better address your concern?**
> >
> > [1] Towards Robust Mathematical Reasoning
> >
> > [2] Length-Controlled AlpacaEval: A Simple Way to Debias Automatic Evaluators
> >
> > ---
> >
> > > **Q4. Have you tested if the attacked prompts still effectively trigger model safety violations across more diverse models (e.g., closed-source ones like Claude 3)? And how do you ensure the attack tactics don’t introduce scenario-specific biases (e.g., overusing "novel writing" in the Child scenario)?**
> > >
> >
> > A4. Sure.
> >
> > **(a) “Have you tested if the attacked prompts still effectively trigger model safety violations across more diverse models?”**
> >
> > Yes. In addition to the Gemini-2.5-flash case study in Appendix G.7 and Figs. 44, 45 and 46, **we included a new analysis on Claude-4.5-haiku in the revised version. As shown in Appendix G.7 and Figs. 47, 48 and 49**, the attack enhancement transforms an unsafe request into a seemingly benign fictional setup, masking the harmful intent while keeping it recognizable. Claude-4.5-haiku behaves safely on the raw prompt, but after enhancement it produces a detailed and technically replicable account of exploiting surveillance blind spots, coordinating distractions, and misusing architectural knowledge. This indicates that narrative reframing can obscure unsafe intent and trigger violations even in strong closed-source models. **More details are provided in Appendix G.7.**
> >
> > **(b) “How do you ensure the attack tactics don’t introduce scenario-specific biases (e.g., overusing "novel writing" in the Child scenario)”**
> >
> > As noted in Appendix D.3 and in WildTeaming [1], the attack method is model-agnostic and black-box, rewriting unsafe prompts using tactics mined from large-scale, real-world user-chatbot logs. WildTeaming identifies over **5.7K unique clusters** of jailbreak behaviors and composes multiple tactics per prompt (e.g., role-play, format shifts, legal framing), preventing any single style such as “novel writing” from dominating a scenario. By sampling across this broad taxonomy and limiting repeated use of the same category, each scenario receives a **balanced mix** of tactics rather than scenario-specific patterns. This reduces bias and supports a more robust evaluation of model vulnerabilities.
> >
> > [1] WildTeaming at Scale: From In-the-Wild Jailbreaks to (Adversarially) Safer Language Models

---

> > > ### Author Response · Authors · 2025-11-20
> > > **Response to Reviewer vrxC (3 / 4)**
> > >
> > > > **Q5. Qwen3-32B-thinking is proposed as a cost-effective evaluator alternative to GPT-4.1, but its scores are slightly lower. For scenarios with strict safety requirements (e.g., Biochem’s dual-use risks), have you validated if this score gap leads to misjudgments of critical safety violations?**
> > > >
> > >
> > > A5. Thanks for your suggestion. In Appendix G.6, we have analyzed the cross-evaluator correlation between GPT-4.1 and Qwen3-32B-thinking through evaluation results, rank correlation, and rank–rank visualizations. As noted in Sec. 4.1 (lines 314 to 317), Qwen3-32B-thinking serves as a cost-effective and locally deployable option during development, while the stronger GPT-4.1 is used for the primary evaluation.
> > >
> > > We also agree that further exploring consistency on safety violation judgments is important. **To address your concern, we provide both (a) statistical evidence and (b) detailed case study in Appendix G.6 of the revised version.** The following provides a brief overview:
> > >
> > > **(a)** The following table shows **statistical evidence across all 33 models and 5 scenarios**. The first row reports the percentage for which GPT-4.1 and Qwen3-32B-thinking assign exactly the same safety score. The second row shows the average proportion of individual safety-specs where their judgements differ. As shown below, the two evaluators match on about 87% of safety scores, and the spec-level mismatch ratios remain in the low single digits (2.4~7.5%) with an overall average of 4.4%. These results indicate that disagreements on specific safety requirements are uncommon even when both evaluators review long checklists.
> > >
> > > |Scenario|Biochem|Child|Code|Health|Travel|Avg|
> > > |-|-|-|-|-|-|-|
> > > |Safety score equal rate (%)|88.3|86.8|85.5|85.1|88.8|86.9|
> > > |Spec judgement mismatch ratio (%)|4.1|2.4|7.5|4.3|3.9|4.4|
> > >
> > > **(b)** We also provide a **case study in Appendix G.6 and Fig. 56 that highlights a point of divergence between the GPT-4.1 and Qwen3-32B-thinking evaluators**. The example is a rare marginal case where the privacy risk is theoretically possible but not actually instantiated in the assistant’s response, leading the two evaluators to differ in strictness. These tendencies may cause small shifts in absolute scores, but as shown in the results and analysis in Appendix G.6, they do not affect the relative differences between models, and the overall rankings remain stable.
> > >
> > > **Please refer to Appendix G.6 in the revised version for more details.**
> > >
> > > ---
> > >
> > > > **Q6. Have you explored why combining Step 1 and Step 2 comes closest to full ALIGN3, and if adjusting the order of steps (e.g., safety first) would yield better results?**
> > > >
> > >
> > > A6. Sure.
> > >
> > > **(a) “Have you explored why combining Step 1 and Step 2 comes closest to full Align3”?**
> > >
> > > Align3 is a simple TTD method that sequentially introduces the behavioral spec (Step 1), the safety spec (Step 2), and a final holistic audit (Step 3) into the reasoning. From a prompt-engineering perspective, using Step 1 and Step 2 together creates a coherent reasoning flow in which the model first aligns its initial plan with the behavioral requirements and then adjusts that plan to satisfy the safety constraints. This already forms a complete guided thought process, and Step 3 further strengthens the model’s understanding by reviewing the full chain. This explains why the combination of Step 1 and Step 2 comes closest to the full Align3.
> > >
> > > Other ablation settings omit either Step 1 or Step 2. Even with Step 3 included, the model does not independently reason about one of the key specification types, so a performance gap relative to the full Align3 is natural.
> > >
> > > **(b) “if adjusting the order of steps (e.g., safety first) would yield better results?”**
> > >
> > > **In Appendix G.1 and Tab. 4 of the revised version, we add experiments testing different prompt orders.** The results are also shown below:
> > >
> > > |Method|Safety|Behavior|SAR|
> > > |-|-|-|-|
> > > |Align3 (ours)|76.40|74.84|62.92|
> > > |2→1→3|76.33|73.58|62.49|
> > > |3→1→2|74.80|73.11|60.61|
> > > |3→2→1|75.60|72.37|60.81|
> > >
> > > Here, 2→1→3 corresponds to the “safety first” variant. We observe that small changes in the ordering produce only minor performance differences, which is sensible since all variants still include the same specifications. Reversing Steps 1 and 2 leads to only a slight shift and remains competitive, whereas placing Step 3 at the beginning consistently results in weaker performance. **These results have been included in Appendix G.1 in the revised version.**

---

> > > > ### Author Response · Authors · 2025-11-20
> > > > **Response to Reviewer vrxC (4 / 4)**
> > > >
> > > > > **Q7. Most models score <65% SAR, with GPT-5-chat leading. Have you analyzed whether GPT-5-chat’s advantage stems from better handling of specific scenario types (e.g., Code’s vulnerability constraints) or a general edge in balancing safety-behavior trade-offs?**
> > > > >
> > > >
> > > > A7. From the results and analysis in Sec. 6 “Analysis across scenarios” and Appendix G.4, we see that GPT-5-chat leads in all five scenarios rather than only in one particular type. This suggests a broad advantage in balancing behavioral and safety specifications rather than a scenario-specific gain.
> > > >
> > > > GPT-5 safe-completions [1] emphasizes producing responses that remain safe while still providing meaningful help, rather than relying on strict refusal policies. In dual-use areas such as Biochem, the model is guided to hold firm safety boundaries while giving only high level, non-operational assistance. [1] also reports reduced failure severity and higher overall helpfulness. Since SpecBench includes both high vulnerability domains such as Code and high dual-use domains such as Biochem, the consistent improvements across all five scenarios point to a general enhancement in managing the safety-helpfulness balance. We therefore attribute GPT-5-chat’s high SAR to this broader capability rather than to specialization in any single scenario.
> > > >
> > > > [1] From Hard Refusals to Safe-Completions: Toward Output-Centric Safety Training

---

### Official Review · Reviewer_hPxS · 2025-10-31

**Soundness:** 3
**Presentation:** 3
**Contribution:** 2
**Rating:** 6
**Confidence:** 3

**Summary:**

In this paper, the authors propose to tackle the timely issue of specification alignment in LLMs, especially across a large variety of fields. First, as a comprehensive dataset seems to be missing in this area, they introduce SpecBench, a set of scenarios, specifications, and well-curated prompts, mixing synthetic and real-world examples as well as unsafe and safe prompts. This benchmark is used to identify and measure gaps in the specification alignment abilities of different LLMs. To quantify performance on this new benchmark, they propose the Specification Alignment Rate (SAR) as a new metric. Second, they introduce a new test-time approach for reasoning models, Align3, that significantly improves specification alignment without the need for additional fine-tuning through a "Propose-Verify-Refine" deliberation loop.

**Strengths:**

-	The construction of SpecBench is a notable contribution of the paper, providing a much-needed, structured benchmark for evaluating the nuanced task of specification alignment. Its inclusion of diverse scenarios, both synthetic and real-world, and its categorization of specifications (e.g., behavioral, stylistic, safety) make it a valuable resource for the community.
-	The proposed Align3 algorithm is simple, intuitive, and shows significant gain in performance. A key advantage is that it is a test-time method, meaning it can improve the alignment of existing, pre-trained models without requiring costly retraining or fine-tuning.

**Weaknesses:**

-	The novelty of the deliberation mechanism could be better situated within the existing literature on self-correction and iterative refinement. While the Propose-Verify-Refine framework is clean, the core idea of using an LLM to critique and improve its own output has been explored in prior work (e.g., Self-Refine), and the paper could benefit from a clearer differentiation.

- The Align3 approach, by its nature, may introduce significant computational and latency overhead. It requires at least three full model inferences (propose, verify, refine) per prompt, which may make it impractical for real-time applications compared to single-pass generation. The trade-off between alignment performance and inference cost is a key consideration that we think could be discussed more.

**Questions:**

1.	In the introduction, you state “SPECBENCH effectively reveals alignment gaps;” could you clarify whether this refers to gaps in performance between different models, or gaps between a model's high performance on general benchmarks (like MMLU) and its lower performance on specificational alignment tasks?
2.	The Align3 process seems to treat all specifications with equal importance. Have you explored weighting specifications based on priority, or a mechanism for handling potentially conflicting specifications (e.g., "be concise" vs. "include all these details")? Perhaps using weight importance for different safety/behaviour instead of a “all-or-none” approach for safety properties and “uniform” importance for behaviour specifications ?

---

> ### Author Response · Authors · 2025-11-20
> **Response to Reviewer hPxS (1 / 2)**
>
> Thanks for the detailed feedback. Our main contribution is introducing **SpecBench**, a new benchmark built around a proposed paradigm of **Specification Alignment**, which **fits well with the goals of the datasets and benchmarks track.**
>
> However, **it seems you may have viewed our proposed method Align3 as the main contribution of the work.** We realized that our previous ordering of the contributions may have caused some misunderstanding about our work. Sorry for the confusion, and we have slightly revised the ordering to make things clearer.
>
> Specifically, our work centers on the new paradigm of Specification Alignment, which extends the notion of specs introduced in OpenAI’s Deliberative Alignment [1] and offers a more fine-grained, scenario-specific view of model spec or policies used by Google, OpenAI, Anthropic, Meta, and others. Align3 is a lightweight test-time scaling method designed to illustrate the effectiveness of test-time deliberation (TTD) like BoN, SelfRefine, MoreThink, and Align3. It is not intended to be the central contribution of the paper.
>
> **Now, you can check the revised abstract and introduction. We slightly adjusted the ordering of contributions, without changing the actual content.** The main contributions are as follows:
>
> 1. We introduce the challenge of **specification alignment** by emphasizing the need to assess LLMs with scenario-specific specifications (spec) that capture both behavioral and safety requirements.
> 2. We release **SpecBench**, the first benchmark to unify behavioral and safety evaluation across 5 scenarios, 103 spec and 1,500 prompts, offering strong diversity and real-world relevance, enabling individuals, institutions, and companies to easily customize their own scenarios and specifications.
> 3. Experiments on diverse instruct and reasoning models with multiple **Test-Time Deliberation (TTD)** methods reveal significant room for improving specification alignment. We observe that SpecBench effectively exposes alignment gaps and that TTD improves alignment.
> 4. Based on previous TTD methods, we further propose **Align3**, a lightweight TTD method that uses hierarchical reflection and revision to reason over safety and behavioral specification boundaries. It advances the safety-helpfulness trade-off frontier with minimal overhead, achieving up to **11.89%** improvement.
>
> [1] Deliberative Alignment: Reasoning Enables Safer Language Models
>
> ---
>
> > **Q1. While the Propose-Verify-Refine framework is clean, the core idea of using an LLM to critique and improve its own output has been explored in prior work (e.g., Self-Refine), and the paper could benefit from a clearer differentiation.**
> >
>
> A1. As clarified above, the specific prompt design of Align3 is not the main focus of our paper. Our emphasize is on the Specification Alignment paradigm, SpecBench, and the overall effectiveness of TTD. Align3 and the other TTD methods together show how test-time deliberation can further support specification alignment. We have slightly adjusted the order of our contributions accordingly to make things clearer.
>
> At the same time, we agree that iterative critique and refinement have been explored in prior work, including SelfRefine, which is also a TTD method and one we have already experimented with. **Our intention with Align3 is not to introduce a fundamentally new TTD framework but to provide a simple and efficient instantiation tailored to specification alignment.** This is why we intentionally designed Align3 with a minimal prompt: the content in Steps 1 and 2 (Fig. 21) is directly adapted from the specification declaration prompt in Fig. 20. Even with such a lightweight design, Align3 still delivers strong performance (see Tab. 2), showing that a simple TTD setup can be effective in this new task setting.

---

> > ### Author Response · Authors · 2025-11-20
> > **Response to Reviewer hPxS (2 / 2)**
> >
> > > **Q2. The Align3 approach, by its nature, may introduce significant computational and latency overhead. Align3 requires at least three full model inferences (propose, verify, refine) per prompt, which may make it impractical for real-time applications compared to single-pass generation. The trade-off between alignment performance and inference cost is a key consideration that we think could be discussed more.**
> > >
> >
> > A2. We believe there may be a misunderstanding about how Align3 operates, and we would like to clarify:
> >
> > 1. **Align3 runs in a single forward pass rather than three separate inferences.** As described in the second paragraph of Sec. 5, *Align3 is a thinking intervention method*, where short prompts are inserted into the model’s ongoing reasoning within a single inference. As shown in Appendix K Fig. 21, these prompts appear at different points in one continuous reasoning sequence, and the model still produces a single chain of thought followed by one final response.
> > 2. **Table 2 already reports both performance and token consumption, showing that Align3 introduces only a small overhead compared to vanilla generation.** Multi-pass TTD methods typically consume dozens of times more tokens because they generate many intermediate reasoning traces. In contrast, Align3 achieves substantial SAR improvement with fewer than 2k tokens, only slightly higher than the 1.3k-1.5k tokens used by vanilla generation. This demonstrates that Align3 is an efficient and practical option for specification alignment, offering strong gains with minimal additional cost.
> > 3. **To avoid further confusion, we added a brief clarification in the second paragraph of Sec. 5 explicitly highlighting that Align3 operates within a single forward pass.** Please refer to Sec. 5 in the revised version.
> >
> > ---
> >
> > > **Q3. In the introduction, you state “SPECBENCH effectively reveals alignment gaps;” could you clarify whether this refers to gaps in performance between different models, or gaps between a model's high performance on general benchmarks (like MMLU) and its lower performance on specificational alignment tasks?**
> > >
> >
> > A3. Yes, this refers to gaps in performance between different models. As noted in Sec. 4.2 lines 325 to 328, “Our SpecBench presents a moderate level of difficulty and reveals clear performance gaps across models.”
> >
> > ---
> >
> > > **Q4. The Align3 process seems to treat all specifications with equal importance. Have you explored weighting specifications based on priority, or a mechanism for handling potentially conflicting specifications (e.g., "be concise" vs. "include all these details")? Perhaps using weight importance for different safety/behaviour instead of a “all-or-none” approach for safety properties and “uniform” importance for behaviour specifications?**
> > >
> >
> > A4. Thanks for the suggestion. In our framework, specifications are provided entirely by the user, which allows full customization of priorities and conflict-resolution rules. **Different users can define priorities or conflict-resolution rules within their specification text, which are themselves a type of specification.** If Align3 were to impose its own weighting or conflict-handling, it would reduce generality and limit performance across broader cases. Besides, we have already included some conditional specifications in SpecBench, for instance the Health and Travel scenario contains several “if-then” behavioral specs that reflect realistic conditional requirements.
> >
> > **To further illustrate how priority can be expressed directly in natural language, Appendix M presents a case study in the Child scenario where we raise the priority of the 5th behavioral spec.** Under the original specification, the model produces stories featuring only one cultural background. After strengthening this specification, the model produces responses that consistently include characters from at least two cultures and highlight their differences, demonstrating that user-stated priorities can effectively guide model behavior. While such behavioral preferences can be flexibly prioritized by the user, safety specifications remain applied in an all-or-none manner to ensure strict adherence to safety spec, since violating any safety requirement may cross critical safety boundaries and must therefore invalidate the response. Overall, these observations show that our specification setting gives users ample flexibility to articulate their own weighting and conflict-handling strategies while maintaining a careful treatment of safety. **Please refer to Appendix M in the revised version.**

---

### Official Review · Reviewer_dFuL · 2025-11-01

**Soundness:** 3
**Presentation:** 3
**Contribution:** 2
**Rating:** 4
**Confidence:** 3

**Summary:**

This paper proposes Align3, a lightweight test-time deliberation method with three phases: behavior-oriented phase for maximizing helpfulness, safety-guided phase to remove risky content, and a spec audit before producing the final answer. To evaluate methods, it also releases SpecBench, a benchmark including 5 application domains with paired safety and behavioral specs per task, with a combined metric SAR that treats safety as a hard constraint and then scores usefulness. Experiments across different LLMs show Align3 advances the safety–usefulness frontier with small token overhead and competitive results versus other TTD baselines.

**Strengths:**

1. The studied problem is interesting. This paper has clear problem formulation.

2. This paper is well-written.

3. The constructed benchmark is useful to the community.

**Weaknesses:**

1. The proposed method needs a largely fixed prompt template/workflow, but prompt structure, and workflow can affect model benign performance; moreover, prompt optimization is an active research area (e.g., automated prompt search/rewriting, domain-specific templates). How Align3 would integrate with such optimization is not discussed in detail. Also, the evaluation of how the proposed method could influence the benign performance of the model is limited to the SpecBench. Broader results on more mainstream benchmarks would help quantify Align3’s effect on non-safety performance.

2. The proposed method is simply prompt based workflow. Thus, the contribution and the novelty of it could be limited.

3. Heavy reliance on LLM judge leaves room for evaluator bias, potentially reducing the quality of the benchmark comparing to the benchmark with ground-truth or verifiable answers.

4. The failure analysis of Align3 and existing methods are not comprehensive enough. More fine-grained analyses and taxonomy of failure modes would clarify when the method breaks and why.

**Questions:**

See Weaknesses.

---

> ### Author Response · Authors · 2025-11-20
> **Response to Reviewer dFuL (1 / 3)**
>
> Thanks for the detailed feedback. Our main contribution is introducing **SpecBench**, a new benchmark built around a proposed paradigm of **Specification Alignment**, which **fits well with the goals of the datasets and benchmarks track.**
>
> However, **it seems you may have viewed our proposed method Align3 as the main contribution of the work.** We realized that our previous ordering of the contributions may have caused some misunderstanding about our work. Sorry for the confusion, and we have slightly revised the ordering to make things clearer.
>
> Specifically, our work centers on the new paradigm of Specification Alignment, which extends the notion of specs introduced in OpenAI’s Deliberative Alignment [1] and offers a more fine-grained, scenario-specific view of model spec or policies used by Google, OpenAI, Anthropic, Meta, and others. Align3 is a lightweight test-time scaling method designed to illustrate the effectiveness of test-time deliberation (TTD) like BoN, SelfRefine, MoreThink, and Align3. It is not intended to be the central contribution of the paper.
>
> **Now, you can check the revised abstract and introduction. We slightly adjusted the ordering of contributions, without changing the actual content.** The main contributions are as follows:
>
> 1. We introduce the challenge of **specification alignment** by emphasizing the need to assess LLMs with scenario-specific specifications (spec) that capture both behavioral and safety requirements.
> 2. We release **SpecBench**, the first benchmark to unify behavioral and safety evaluation across 5 scenarios, 103 spec and 1,500 prompts, offering strong diversity and real-world relevance, enabling individuals, institutions, and companies to easily customize their own scenarios and specifications.
> 3. Experiments on diverse instruct and reasoning models with multiple **Test-Time Deliberation (TTD)** methods reveal significant room for improving specification alignment. We observe that SpecBench effectively exposes alignment gaps and that TTD improves alignment.
> 4. Based on previous TTD methods, we further propose **Align3**, a lightweight TTD method that uses hierarchical reflection and revision to reason over safety and behavioral specification boundaries. It advances the safety-helpfulness trade-off frontier with minimal overhead, achieving up to **11.89%** improvement.
>
> [1] Deliberative Alignment: Reasoning Enables Safer Language Models
>
> ---
>
> > **Q1. The proposed method needs a largely fixed prompt template/workflow, but prompt structure, and workflow can affect model benign performance; moreover, prompt optimization is an active research area (e.g., automated prompt search/rewriting, domain-specific templates). How Align3 would integrate with such optimization is not discussed in detail.**
> >
>
> A1. As clarified above, the specific prompt design of Align3 is not the main focus of our paper. Our emphasize is on the Specification Alignment paradigm, SpecBench, and the overall effectiveness of TTD. Align3 and the other TTD methods together show how test-time deliberation can further support specification alignment. We have adjusted the **order** or our contributions accordingly to make things clearer.
>
> That said, the prompt design of Align3 can indeed influence its performance. For this reason, we intentionally used a simple prompt. As you can see, the prompt content in Steps 1 and 2 (Fig. 21) is simply adapted from the specification declaration prompt in Fig. 20. Despite its simplicity, the performance of Align3 remains encouraging (see Tab. 2). Exploring domain-specific templates or prompt optimization is a natural next step for our proposed TTD, but it is beyond the scope of this paper.

---

> > ### Author Response · Authors · 2025-11-20
> > **Response to Reviewer dFuL (2 / 3)**
> >
> > > **Q2. The evaluation of how the proposed method could influence the benign performance of the model is limited to the SpecBench. Broader results on more mainstream benchmarks would help quantify Align3’s effect on non-safety performance.**
> > >
> >
> > A2. Thanks for your valuable suggestions. We fully agree that understanding the impact on benign performance is important. But in our case, Align3 operates only under explicit specification alignment settings, and it does not modify model parameters. Therefore, it does not affect the model’s general abilities or benign performance when no spec is provided.
> >
> > 1. **The main focus of our paper is not the design of Align3.** Our central contributions are the Specification Alignment paradigm, SpecBench, and the demonstrated effectiveness of TTD. Align3 is simply one TTD method we introduce; other TTD such as TPO and MoreThink can also improve performance. Align3 is just a more efficient and direct option among them.
> > 2. **Align3 is designed as a lightweight TTD method specifically for the specification alignment challenge.** To the best of our knowledge, SpecBench is the first benchmark that formulates and evaluates this task, so it cannot be directly tested on previous mainstream benchmarks. Align3 requires an explicit specification to operate, whereas existing mainstream benchmarks do not include such specification-based settings.
> > 3. **Specification Alignment is a new task tailored for scenario-specific requirements.** It differs from benchmarks on scientific QA, instruction following, math, or purely safety-oriented evaluations. For example, ArenaHard, AlpacaEval, and Google’s recent IMOBench primarily evaluate correctness. The differences and limitations of previous instruction-following and safety benchmarks are also discussed in Sec. 7 Related Work.
> >
> > ---
> >
> > > **Q3. The proposed method is simply prompt based workflow. Thus, the contribution and the novelty of it could be limited.**
> > >
> >
> > A3. Thanks for raising this point. We would like to clarify the contribution and novelty from two perspectives:
> >
> > 1. **Our core contributions lie in the Specification Alignment paradigm and SpecBench, where we evaluate 33 models and demonstrate the effectiveness of TTD for alignment. Align3 is simply one of these TTD methods, serving as an efficient and direct option.** Specification Alignment offers a fine-grained and user-customizable view of model specifications, extending the concept introduced in OpenAI’s Deliberative Alignment and refining model spec practices used by Google, OpenAI, Anthropic, Meta and others. This perspective is fundamentally different from prior safety or instruction-following work discussed in Sec. 7. As LLMs move into real-world deployments, scenario-specific responses become increasingly necessary.
> > 2. **SpecBench is not simply a prompt-based workflow. The spec and data construction rely on extensive real-world sources, careful technical processing, and continuous human-in-the-loop review, as detailed in Sec. 3 and Appendix F.** For example, we incorporate model specs from companies like Google and concepts from OpenAI’s Deliberative Alignment to formalize the challenge (Sec. 2); we build representative datasets grounded in real scenarios (Appendix D.1 and D.5); and we conduct sustained human review to ensure data reliability (Appendix D.4 and F). These steps require substantial effort and go far beyond prompt design.

---

> > > ### Author Response · Authors · 2025-11-20
> > > **Response to Reviewer dFuL (3 / 3)**
> > >
> > > > **Q4. Heavy reliance on LLM judge leaves room for evaluator bias, potentially reducing the quality of the benchmark comparing to the benchmark with ground-truth or verifiable answers.**
> > > >
> > >
> > > A4. We completely agree with your concern, but this challenge is indeed shared by many current benchmarks:
> > >
> > > 1. **LLM-as-a-judge is widely adopted across mainstream benchmarks.** Beyond tasks with verifiable answers such as math reasoning, many general-purpose benchmarks (e.g., MT-Bench, ArenaHard [1,2]), safety-oriented benchmarks (e.g., AIR-Bench 2024 [3]), and even Google’s recent IMO-Proof-Bench for math reasoning [4] rely on LLM-as-a-judge. Given the scale of modern evaluation workloads, relying solely on human annotation is infeasible and prohibitively time-consuming and expensive for most research settings.
> > > 2. **We take multiple steps to minimize evaluator bias.** We conduct a consistency study measuring agreement with human experts and with an additional open-source model (Qwen3-32B-thinking). The correlation remains high and trends are consistent. Our human evaluation study (Appendix F), following prior work [3], shows a kappa of 0.84 between three human experts and GPT-4.1. The strong agreement and consistent rankings between GPT-4.1 and Qwen3-32B-thinking (Sec. 4.1 and Appendix G.6) also provide a cost-effective and locally deployable alternative for development.
> > > 3. **To further address your concern, we additionally test alternative evaluators.** We incorporate the newest Claude-4.5-Haiku and DeepSeek-V3.2-Exp as evaluators for a subset of models and find that the ranking results remain largely stable. We also compute Kendall’s coefficient of concordance across all evaluators, obtaining a near-perfect $W=0.9948$ with $\chi^2=43.77$ (df = 11) and $p=7.97\times10^{-6}$, confirming extremely strong overall agreement. **Please refer to Sec. 6 in the revised version for more details.**
> > >
> > > [1] Judging LLM-as-a-Judge with MT-Bench and Chatbot Arena.
> > >
> > > [2] Length-Controlled AlpacaEval: A Simple Way to Debias Automatic Evaluators.
> > >
> > > [3] AIR-Bench 2024: A Safety Benchmark Based on Risk Categories from Regulations and Policies.
> > >
> > > [4] Towards Robust Mathematical Reasoning
> > >
> > > ---
> > >
> > > > **Q5. The failure analysis of Align3 and existing methods are not comprehensive enough. More fine-grained analyses and taxonomy of failure modes would clarify when the method breaks and why.**
> > > >
> > >
> > > A5. We have provided detailed case studies in Appendix G.5, including:
> > >
> > > - Figs. 33, 34, 35: safety–behavior trade-off cases
> > > - Figs. 36, 37, 43: successful cases
> > > - Figs. 38, 39, 40: alignment gap comparisons
> > > - Figs. 41, 42: two representative failure cases
> > >
> > > These already cover a wide range of examples to help you understand our work.
> > >
> > > **Nevertheless, the two representative failure cases were not broad enough, and we agree that more examples would help. To address this, we add six additional failure cases in Appendix G.5, including Figs. 50, 51, 52, 53, 54, and 55.** They are failures from different models and different test-time deliberation methods, each with detailed analysis. Please refer to the updated Appendix G.5 in the revised version.
> > >
> > > ---
> > >
> > > **We sincerely appreciate your time and effort in reviewing our submission and our response, and we are of course happy to address any additional questions or concerns you may have.**

---

> > > > ### Author Response · Authors · 2025-11-25
> > > > **Follow-up on Our Rebuttal Submission**
> > > >
> > > > Dear Reviewer,
> > > >
> > > > Since it has been five days since we submitted our rebuttal, we wanted to kindly follow up in case you have any further thoughts or questions regarding our submission.
> > > >
> > > > We truly hope our responses were helpful in addressing your comments. If there’s anything further you’d like to discuss or clarify, we would be very grateful to hear your feedback before the discussion period ends.
> > > >
> > > > Warm regards,
> > > >
> > > > The authors

---

> > > > > ### Author Response · Authors · 2025-11-27
> > > > > **Kind Follow-up on the Discussion of Our Rebuttal**
> > > > >
> > > > > Dear Reviewer dFuL,
> > > > >
> > > > > Thank you for reviewing our work and providing feedback. We submitted our response to your comments a few days ago. Since the rebuttal window is closing, we would greatly appreciate it if you could share any thoughts so that we can continue the discussion and address any concerns you may have.
> > > > >
> > > > > In summary, to address your points, we have added new experiments and discussion as shown below, which are also integrated into the updated version.
> > > > >
> > > > > First, **we clarify that the core contribution of our work lies in the Specification Alignment paradigm, the construction of SpecBench, and our finding that Test-Time Deliberation (TTD) is effective, which aligns well with the goals of the datasets and benchmarks track, rather than in our simple TTD method Align3.** To avoid misunderstanding, we have slightly adjusted the ordering of contributions in the abstract and introduction.
> > > > >
> > > > > Based on these points, we further clarify the role of Align3 **(Q1, Q2, Q3)**. We explain that Align3 is specifically designed for the Specification Alignment challenge **(Q2).** We emphasize that our main contributions lie in the Specification Alignment paradigm, the construction of SpecBench, and our finding that TTD enhances alignment. The novelty of the work is therefore not diminished (**Q3**).
> > > > >
> > > > > Secondly, for **Q4**, we take multiple steps to minimize evaluator bias and additionally include the newest evaluators (**DeepSeek-V3.2-Exp and Claude-4.5-haiku, Sec. 6**), confirming that different evaluators exhibit high correlation.
> > > > >
> > > > > Finally, for **Q5**, we add 6 more failure cases from different models and methods to give you a clearer view of when these approaches may fail (**Appendix G.5 Figs. 50, 51, 52, 53, 54, and 55**).
> > > > >
> > > > > We would appreciate it if you could kindly take a look at our response to your comments. If you have any further questions, we are happy to address them!
> > > > >
> > > > > Best regards,
> > > > >
> > > > > The authors

---

### Author Response · Authors · 2025-12-03
**Overview of Our Response and Key Points**

Dear AC,

Thank you for taking the time to review our rebuttal, and we sincerely thank all reviewers for their time and efforts. Their comments have been very helpful for refining our work.

Across the reviews, we identified and clarified **`two core misunderstandings`**, including one complete misinterpretation of the paper despite clear statements in the main text. In addition, we address **`three key concerns`** regarding **evaluator reliability, effectiveness, and case robustness**.

---

## **Two Core Misunderstandings**

> ***Misinterpreting the Main Contributions* [*Reviewer dFuL.Q1/2/3, hPxS.Q1*]**
>

Reviewers *dFuL and hPxS* treated Align3 as the main contribution, which directly triggered concerns regarding novelty and relation to prior work.

However, as stated repeatedly, the **central contribution** is the Specification Alignment paradigm, SpecBench, and the TTD findings, which fits well with the goals of the ***datasets and benchmarks primary area***. Specification Alignment extends the notion of specs introduced in OpenAI’s Deliberative Alignment and offers a more fine-grained, scenario-specific view of model spec or policies used by *Google, OpenAI, Anthropic, Meta, and others*. Align3 is simply a lightweight, single-pass TTD to illustrate the effectiveness of TTD methods (e.g., BoN, SelfRefine, MoreThink. Sec. 5).

To make things clearer, we slightly adjusted the ordering of contributions (abstract, introduction) without changing any substantive content.

> ***Misunderstanding the Specification Alignment Paradigm* [*Reviewer 8wVs.Q2/3/4*]**
>

Reviewer *8wVs* treated Specification Alignment as requiring a comprehensive or universally defined set of rules. This led to concerns about spec completeness and evolving requirements.

However, as described in Sec. 1-3, a specification is a **direct expression of user intent, not an attempt to enumerate all possible rules**. Specifications are inherently scenario-specific, user-defined, subjective, and easily updated. Our framework naturally supports evolving or changing specifications because the model aligns to the specification given at inference time.

To make things clearer, we add high-level instructions for constructing new scenarios (Appendix H) and a simulation in which we update the specifications and re-evaluate the models, showing that rankings remain consistent under evolving specifications (Appendix L).

---

## **Three Key Concerns**

> ***Evaluator Reliability* [*Reviewer dFuL.Q4, vrxC.Q1/5, 8wVs.Q1*]**
>

We have already taken multiple steps in the original version to minimize evaluator bias (Appendix F human study, Sec. 4.1 and Appendix G.6 consistency analysis). We further include new evaluators (DeepSeek-V3.2-Exp and Claude-4.5-Haiku, Sec. 6), which show strong correlation (Kendall’s W = 0.9948), confirming stable rankings and addressing all evaluator-related concerns.

> **Effectiveness [*Reviewer hPxS.Q2/4, vrxC.Q3/6*]**
>

We address the incorrect misunderstanding that Align3 “introduces significant overhead” and is “impractical”, clarifying that the original version already emphasized its efficiency repeatedly (e.g., 1550 → 1832 tokens) [*hPxS.Q2*]. We show that user-defined specifications naturally encode priority or conflict-handling, and provide a case study to illustrate this [*hPxS.Q4*]. We also clarify that Align3 is test-time only without long-term effects [*vrxC.Q3*] and include additional ablations demonstrating its effectiveness [*vrxC.Q6*].

> **Case Robustness [*Reviewer dFuL.Q5, vrxC.Q4, 8wVs.Q3*]**
>

We provide 6 more failure cases to better illustrate alignment gaps [*dFuL.Q5*]. We include expanded attack case study (Claude-4.5-Haiku) with analysis showing the effectiveness of attack enhancement [*vrxC.Q4*]. We also simulate the dynamic and evolving nature of our spec settings, confirming robustness under spec updates [*8wVs.Q*3].

---

> ### Author Response · Authors · 2025-12-03
> **[Further Details] Reviewer Concerns and Responses**
>
> We sincerely thank the newly assigned AC, the originally assigned AC, and all reviewers for their time and efforts. The review process has helped us significantly improve our paper. We provide this summary to clearly present each reviewer’s questions and our responses.
>
> ---
>
> ## **Reviewer dFuL (Rating 4, Conf 3. No Reply)**
>
> > **Q1, Q3. The proposed method Align3 is simply a prompt workflow, reducing the novelty, and its relation to prior prompt workflows is not fully discussed.**
> >
>
> These arise from **the first misunderstanding**. We address them by emphasizing that our main contribution lies in the Specification Alignment paradigm, SpecBench, and TTD, rather than in the simple TTD method Align3. We consider diverse TTD methods to enhance specification alignment, including prompt workflows, and our Align3 is simply one such method. Its simplicity does not diminish the novelty of our main contributions.
>
> > **Q2. Evaluation of Align3 on other tasks is limited.**
> >
>
> This also arises from **the first misunderstanding**. We clarify that Align3 is not our main focus. It is tailored for our specification alignment and spec-dependent, and SpecBench is the first benchmark to formulate this task.
>
> > **Q4. Evaluator reliability on LLM-as-a-judge.**
> >
>
> We clarify that most existing benchmarks also rely on LLM-as-a-judge and that pure human evaluation is costly and impractical. We have taken multiple steps to support reliability and minimize evaluation bias (human study, consistency analysis), and we add new evaluators (Claude-4.5-Haiku, DeepSeek-V3.2-Exp), which show strong correlation (Kendall’s W = 0.9948).
>
> > **Q5. More case studies.**
> >
>
> We agree that more examples would help and add six cases (Appendix G.5 Figs. 50, 51, 52, 53, 54, 55) to make this clearer.
>
> ---
>
> ## **Reviewer hPxS (Rating 6, Conf 3. No Reply)**
>
> > **Q1. The core idea of using an LLM to improve its own output (Align3) has been explored in prior work.**
> >
>
> Same concern as addressed in [*Reviewer dFuL.Q1*].
>
> > **Q2. Align3 “introduces significant overhead” and is “impractical”.**
> >
>
> This is an incorrect misunderstanding. We repeatedly show in Sec. 5 (TTD design and token analysis) that Align3 adds ≤ 400 tokens in practice (e.g., 1550 → 1832 tokens), demonstrating that it is highly efficient rather than “impractical”.
>
> > **Q3. What does “alignment gaps” refer to.**
> >
>
> We clarify that “alignment gaps” refer to performance differences across models.
>
> > **Q4. Whether specifications can support priority or conflict handling.**
> >
>
> We clarify that user-defined specifications naturally encode priority and conflict-handling, and we add a case study illustrating this flexibility (Appendix M).
>
> ---
>
> ## **Reviewer vrxC (Rating 6, Conf 4. No Reply)**
>
> > **Q1. Evaluator reliability and evaluation bias.**
> >
>
> Same concern as addressed in [*Reviewer dFuL.Q4*].
>
> > **Q2. More discussion on low-resource languages or niche domains.**
> >
>
> We have provided a construction guideline and additionally include high-level instructions (Sec. 3, Appendix D, H) that can be easily extended to low-resource languages or niche domains.
>
> > **Q3. Align3’s long term performance.**
> >
>
> We clarify that Align3 is test-time only and spec-dependent, with no long-term or “over-alignment” effects.
>
> > **Q4. Effectiveness and diversity of attack enhancement.**
> >
>
> We test attacks on Claude-4.5-Haiku and add new examples (Appendix G.7, Figs. 47, 48, 49), demonstrating both validity and diversity.
>
> > **Q5. Safety agreement across two evaluators.**
> >
>
> Building on the three original consistency analyses, we further include experiments showing stable rankings and high agreement (Appendix G.6).
>
> > **Q6. More ablations (reverse-step ordering) on Align3.**
> >
>
> In addition to the step ablations, we further add reverse-step ordering ablations on Align3 (Appendix G.1), confirming the effectiveness of our settings.
>
> > **Q7. Why GPT-5-chat leads.**
> >
>
> We restate that GPT-5-chat leads across all five scenarios (Sec. 6, Appendix G.4), indicating broad and consistent strength.
>
> ---
>
> ## **Reviewer 8wVs (Rating 4, Conf 4. No Reply)**
>
> > **Q1. Evaluation bias on LLM-as-a-judge.**
> >
>
> Same concern as addressed in [*Reviewer dFuL.Q4*].
>
> > **Q2, Q4. Lack of a systematic approach for creating a complete set of specs.**
> >
>
> This concern arises from **the second misunderstanding**. We clarify that a specification is a direct expression of user intent, not an attempt to enumerate all possible rules. Specifications are inherently scenario-specific, user-defined, subjective, and easily updated through our flexible guidelines (Appendix D.4, D.5, H).
>
> > **Q3. Limited treatment of dynamic and evolving specs.**
> >
>
> This also arises from the **second misunderstanding**. To further validate this, we add an update-simulation experiment with results showing consistent rankings (Appendix L).

---

### Meta-Review · Area_Chair_6dyi · 2026-01-09

**Summary:**

Despite the authors' detailed rebuttal,  several of the reviewers' original concerns remain not fully resolved hence the paper can't be accepted

1.Reviewers dFuL and hPxS correctly identified that the proposed method, Align3, lacks significant novelty, describing it as a "simply prompt based workflow" and noting that similar iterative refinement ideas (like Self-Refine) have been explored previously.

In response, the authors attempted to pivot the paper's primary contribution away from the method and toward the SpecBench benchmark and the "Specification Alignment paradigm". If this is simply a misunderstanding of the reviewers, I highly encourage the authors to imprtove writing of the paper in the next submission


2. The Validity of the Benchmark Construction
Reviewer 8wVs raised a fundamental issue regarding the methodology for creating specifications, noting it relies on a manual process that is "difficult to scale and validate" and lacks a "systematic framework" or "stopping criteria" for creating complete specs.

The authors countered by arguing that specifications are "inherently subjective" and "user-centered," implying that a systematic definition of completeness is unnecessary because specs are just "statements of user intent".

This is a philosophical dodge rather than a technical resolution. For a scientific benchmark (SpecBench) to be robust, it must have a reproducible standard for what constitutes a "complete" or "fair" specification.



3. Multiple reviewers (dFuL, vrxC, 8wVs) flagged the heavy reliance on GPT-4.1 as a judge, citing potential evaluator bias.

The authors rebutted this by showing high correlation between GPT-4.1, human annotators (on a small subset), and other models like Qwen and Claude.

High correlation between LLMs does not prove ground truth; it often proves shared bias. Most RLHF (Reinforcement Learning from Human Feedback) models share similar training signals regarding safety and refusal. * Result: If GPT-4, Claude, and Qwen all agree to penalize a specific nuance of "helpfulness" in favor of "safety" due to their safety-tuning, the benchmark might simply be measuring "conformity to standard safety tuning" rather than true specification alignment. The human study (300 samples) is likely too small to fully exonerate the metric from this systemic bias given the scale of the benchmark (1,500 prompts).

4. Computational Practicality Reviewer hPxS raised concerns about the inference overhead of Align3, noting it requires multiple forward passes or reasoning steps.

While the authors argue that Align3 is "lightweight" because it happens in a single pass of "thinking," it still increases the token count and latency compared to standard inference. In real-time applications, any latency increase is a friction point.
Therefore, The trade-off between the marginal gains in alignment (approx. 11% improvement in SAR) versus the added latency/cost of the "thinking" process is a valid engineering concern that the "lightweight" label glosses over.

**Reviewer Concerns:**

see above

**Reviewer Scores:**

see above

---

### Decision · Program_Chairs · 2026-01-26

Reject